# Investigation of the Antifungal and Anticancer Effects of the Novel Synthesized Thiazolidinedione by Ion-Conductance Microscopy

**DOI:** 10.3390/cells12121666

**Published:** 2023-06-19

**Authors:** Nikita Savin, Alexander Erofeev, Roman Timoshenko, Alexander Vaneev, Anastasiia Garanina, Sergey Salikhov, Natalia Grammatikova, Igor Levshin, Yuri Korchev, Petr Gorelkin

**Affiliations:** 1Research Laboratory of Biophysics, National University of Science and Technology MISiS, Moscow 119049, Russia; nsavin99@mail.ru (N.S.);; 2G. F. Gauze Research Institute for New Antibiotics, Moscow 119021, Russialevshin@panavir.ru (I.L.); 3Faculty of Medicine, Imperial College London, London SW7 2DD, UK

**Keywords:** SICM, cell stiffness, ROS level, thiazolidinedione, antifungal, anticancer, *Candida*, HEK239, PC3

## Abstract

In connection with the emergence of new pathogenic strains of *Candida*, the search for more effective antifungal drugs becomes a challenge. Part of the preclinical trials of such drugs can be carried out using the innovative ion-conductance microscopy (ICM) method, whose unique characteristics make it possible to study the biophysical characteristics of biological objects with high accuracy and low invasiveness. We conducted a study of a novel synthesized thiazolidinedione’s antimicrobial (for *Candida* spp.) and anticancer properties (on samples of the human prostate cell line PC3), and its drug toxicity (on a sample of the human kidney cell line HEK293). We used a scanning ion-conductance microscope (SICM) to obtain the topography and mechanical properties of cells and an amperometric method using Pt-nanoelectrodes to register reactive oxygen species (ROS) expression. All data and results are obtained and presented for the first time.

## 1. Introduction

The emergence of new pathogenic strains of the fungus and their resistance to clinical antifungal agents are raising public health concerns around the world. According to a recent WHO report [1], cases of invasive fungal diseases are increasing, due to the weakening of immune systems in some populations, due to the widespread availability of modern treatments such as chemotherapy and cancer immunotherapy, and epidemics of viral infections of the respiratory tract, such as influenza and coronavirus disease (COVID-19). Indeed, candidiasis is one of the causes of death in patients with COVID-19 [2]. The global health threat of pathogenic fungi is caused not only by the emergence of antifungal-resistant yeast strains, but also by the lack of available diagnostics [3,4] and the high cost of antifungal drugs that are often highly toxic [5,6]. In this regard, it is necessary to develop and study new, highly effective and inexpensive antifungals, with low-toxicity, for the treatment of candidiasis.

Drug substance availability and its pharmacological activity are determined by physicochemical parameters, such as solubility in biological fluids and permeability through intestinal epithelium barriers [7,8]. One of the intensively developing directions in the search for new highly active antifungal agents is the creation of hybrid molecules [9,10,11]. A significant part of commercial preparations and newly synthesized substances are molecules with a triazole fragment as one of the constituent parts of the hybrid molecule [12]. A group of researchers led by Dr. I. Levshin created active hybrid compounds, with the inclusion of a triazole part, a pharmacophore group based on a thiazolidine fragment [13,14]. The resulting hybrid amide derivatives are considered potential antifungal drugs with a wide spectrum of fungicidal activity. One of the promising compounds, (5Z)-5-[(4-chlorophenyl)methylidene]-3-(2-{4-[2-(2,4-difluorophenyl)-2-hydroxy-3-(1,2,4-triazol-1-yl)propyl]piperazin-1-yl}-2-oxoethyl)-1,3-thiazolidine-2,4-dione) (L-173) was found to have some significant physicochemical properties, when its solubility in buffer solutions (pH 1.2, 2.0 and 7.4) in various temperature ranges was studied and its thermodynamic solubility and transfer parameters were determined [15]. Additionally, in order to increase solubility, complexes of L-173 with various cyclodextrins were studied, and their distribution between biological tissues and penetration through biological membranes were evaluated [16]. Furthering this line of study, the antifungal and anticancer activities, as well as the toxicity of this particular substance (L-173) were investigated in the present study.

It is worth noting that antifungal drugs (e.g., itraconazole, econazole, and ketoconazole) are also capable of exhibiting anticancer activity [17,18,19,20,21]. This combination of properties may contribute to the development of new methods for the treatment of cancer patients, as well as those with fungal diseases, as noted earlier. Thus, the study of the pharmacological effects of 4-thiazolidinones shows that 5-en-2-(imino)amino-4-thiazolidinones are one of the promising groups of compounds with anti-inflammatory, antibacterial, and antitumor activity [22,23]. In addition, moderate antitumor activity of 4-substituted aminothiazol-2(5H)-one derivatives has been reported. 

In addition to interfering with ergosterol biosynthesis, azole (miconazole) drugs induce the accumulation of reactive oxygen species (ROS) [24,25,26,27,28]. In the case of miconazole, a correlation was found between the amount of ROS induction and its fungicidal activity against *Candida* spp., such as *Candida albicans* and *Candida glabrata* [28]. In this regard, it was demonstrated that the suppression of miconazole-induced ROS accumulation with the antioxidant pyrrolidine dithiocarbamate, led to a significant decrease in the antifungal activity of miconazole [28]. In addition, it was previously shown that an increase in ROS in *C. albicans* cells under the action of miconazole can be explained, in part, by the inhibition of catalase and peroxidase, which are important enzymes in the breakdown of peroxide radicals and hydrogen peroxide [27]. To the best of our knowledge, the ability of fluconazole and compounds from the azole group to induce ROS in healthy and cancerous cells has never been studied before.

In 2022, WHO presented an updated list of priority fungal pathogens against which the development of new antifungal medicines is recommended [1]. In this work, the following pathogens from the proposed list were studied as samples: *Candida albicans* (critical priority group), *Candida parapsilosis* (high priority group), and *Candida krusei* (middle priority group). Previously, we developed and tested a method for obtaining the topography and mechanical properties of yeast by scanning ion-conductance microscopy (SICM), using the example of the effect of commercial preparations on the surface structure of *Candida* [29].

Scanning ion-conductance microscopy is a multifunctional technology and a very promising method for studying living cells [30,31,32]. Due to the use of non-contact probing and, consequently, minimal surface deformation, the method is practically non-invasive [33,34,35]. SICM has been successfully used to obtain the topography of samples and conduct non-contact measurements of local mechanical properties, which makes this method optimal for studying the effect of drugs on living cells. Moreover, the use of the method for intracellular detection of ROS using platinum nanoelectrodes will complement the picture of the processes taking place in cells and will make it possible to determine the toxicity of antifungal drugs to healthy embryonic kidney cells and prostate cancer cells [36,37].

In connection with the above, the purpose of this work is to confirm the antitumor activity of a drug of the azole group and to study the cytostatic efficacy of thiazolidinediones in the development of future drugs for tumor cells, paying attention to morphology, mechanical properties of cells, and the level of intracellular ROS. Moreover, an analysis of the toxicity of antifungal drugs on a healthy cell line was carried out.

## 2. Materials and Methods

### 2.1. Substrate Modification and Yeast Cell Preparation

The study was conducted on *Candida* strains obtained from the working collection of the Gause Institute of New Antibiotics: clinical isolates of *C. albicans* 8R, *C. krusei* 432M, resistant to fluconazole, and the control strain *C. parapsilosis* ATCC 22019. To obtain high-quality, high-resolution images and rescan the area without noticeable cell drift, it was advisable to immobilize the cells on the substrate. For this, the surface of the Petri dish was modified. To avoid the drift of microorganisms under the action of small vibrations in an aqueous medium, the surface of the Petri dishes was coated with SYLGARD™ 184 silicone elastomer, after which the dishes were placed in a dry oven for an hour at a temperature of T = 60 °C. The Petri dishes were then thoroughly rinsed with distilled water and dried to remove any excess liquid. A 48 h yeast culture grown on Sabouraud Dextrose Agar was suspended in Hank’s solution to a density of 1 × 10^7^ CFU/mL. The cell concentration was determined by counting in a Goryaev chamber. A small drop of the prepared suspension was applied to the modified substrate. The time for cell sedimentation and their immobilization on the surface was three hours. After that, Hank’s balanced salt solution or Hank’s working solution with the drug was poured into the Petri dish for the control group and the experimental group, respectively. Samples of drug powders were dissolved in dimethyl sulfoxide with a purity of 99.9%, to a concentration of 0.1 mg/mL, and then brought to the test concentrations in 1 mL of Hank’s solution. The samples were incubated with the preparation during the day. After that, the Petri dishes with the samples were preliminarily washed with saline, filled with 2 mL of Hank’s solution, and placed on the scanning platform.

### 2.2. Cell Lines

PC-3 (human prostate cancer cells) and HEK293 (human embryonic kidney) cells were purchased from the American Type Culture Collection (ATCC, Manassas, VA, USA). PC-3 cells were cultured in RPMI-1640 medium (Gibco, Inchinnan, UK) supplemented with 10% FBS (Gibco, Inchinnan, UK), 2 mM L-glutamine (Gibco, Inchinnan, UK), and 1% RPMI vitamin solution (Sigma, Saint Louis, MO, USA). HEK293 cells were cultured in DMEM/F-12 medium (Gibco, Inchinnan, UK) supplemented with 10% FBS and 2 mM L-glutamine. Cells were maintained at 37 °C in a humidified incubator (Sanyo, Osaka, Japan) supplied with 5% CO_2_.

### 2.3. Cytotoxicity Assay

Cells were seeded in the 96-well plates (Corning) at a concentration of 4 000 cells per well for PC-3 culture and 6 500 cells per well for the HEK293 line. An automated cell counter MOXI was used to calculate the cells. After 1 day, serial dilutions of fluconazole or L173 in DMSO were added to the cells. Culture medium with DMSO at concentrations corresponding to those added with the experimental samples was used as a control. The cell medium was used as a negative control, while 30% DMSO diluted in the medium was used as a positive control. Cells were then incubated for 72 h at 37 °C and 5% CO_2_. Later, the culture medium from each well was removed, and 20 μL of MTS reagent (CellTiter 96 AQueous Non-Radioactive Cell Proliferation Assay, Promega) were added to each well with 100 μL of new culture medium. After 4 h of incubation at 37 °C in darkness, the absorbance of the solution was measured at 490 nm wavelength using a Thermo Scientific Multiskan GO spectrometer. Cell viability was calculated as a percent compared to cells incubated in the culture medium. The absorbance of MTS reagent in culture medium without cells was taken as zero. The MTS assay revealed 100% cell death after incubation with 30% DMSO. DMSO at concentrations corresponding to those added with the samples did not cause cell viability to decrease. Experiments were performed in triplicate. 

The data were analyzed using the *t*-test in GraphPad Prism 9 software. *p* values less than 0.05 were considered significant.

### 2.4. Scanning Ion-Conductance Microscopy

Non-contact topographic imaging and low-stress mechanical property measurements were performed using SICM manufactured by ICAPPIC (ICAPPIC Limited, Lonon, UK). SICM by ICAPPIC was mounted on an inverted optical microscope Eclipse Ti-2 (Nikon, Tokyo, Japan) and covered with a Faraday cage for electrical noise shielding. The ICAPPIC Universal Controller and Piezo Control System (ICAPPIC Limited, Lonon, United Kingdom) was used for piezo positioning and feedback control. A MultiClamp 700B amplifier (Molecular Devices, San Jose, CA, USA) was used for ion current monitoring. Installation settings were set as follows: Hop amplitude 2000 nm, fall rate 5 nm/ms, Delay 4000 µs, Length 4 ms, drop duration 20 samples, Hop rise rate 800 nm/ms, Brake boots On, Pre-scan sqr size 4.286 µm, Pre-scan hop 5000 nm, Velocity 25 µm/s.

Nanopipettes were fabricated using a P-2000 laser puller (Sutter Instruments, Novato, CA, USA) from borosilicate glass capillaries (O.D. 1.2 mm, I.D. 0.90 mm, 7.5 cm length). The capillary is placed in the clamps of the adjustable platform. A laser beam is emitted, heating the workpiece in the middle. At the same time, the workpiece is pulled in opposite directions along the horizontal axis until the capillary breaks into two nanopipettes. Puller program for obtaining nanopipettes with a radius of 45 nm: Heat 310, Fil 3, Vel 30, Del 160, Pul 0, Heat 330, Fil 3, Vel 25, Del 160, and Pul 200. Puller program for obtaining nanopipettes with a radius in the range of 20 to 35 nm: Heat 350, Fil 3, Vel 35, Del 160, Pul 0, Heat 330, Fil 3, Vel 50, Del 160, and Pul 220.

Human prostate adenocarcinoma PC3 and human embryonic kidney HEK 293 cell lines (ATCC, Manassas, VA, USA) were cultured in Dulbecco’s modified Eagle’s medium/Nutrient Mixture F-12 (Gibco, Grand Island, NE, USA) containing 1% GlutaMAX (Gibco, USA), 1% Penicillin-Streptomycin (Gibco, USA), 10% fetal bovine serum (Gibco, USA) and maintained in a humidified incubator at 5% CO_2_ and 37 °C. Cell cultures were tested for the absence of mycoplasma. One day before the experiment 2 × 10^5^ of PC3 and HEK293 cells were seeded in 35 mm Petri dishes. On the next day, the culture dishes were refilled with 2 mL of warm culture medium containing 66 µM of fluconazole and 34 µM of L-173. Untreated cells were used as a control. After 40 min of incubation, cells were gently washed with warm Hank’s balanced solution (PanEco, Moscow, Russia) three times and used for SICM measurements.

For mammalian cells, topography and mechanical property measurements were obtained in HBSS [35,38], with the first setpoint set at 0.5% ion current drop (for topography).To obtain maps of the Young’s modulus, the second and third setpoints were set at 1%, 2% ion current drop, respectively. Borosilicate probes with a radius of 45 nm were used. The topography of yeast cells immobilized on a layer of SYLGARD™ 184-silicone elastomer and in HBSS was scanned according to a previously developed protocol [29,39]. The exceptions to the protocol were the relatively low setpoint of the first setpoint of 0.2% ion current drop and the nanopipette radius in the range of 20 to 35 nm. The nanopipette radius and mechanical characteristics are calculated using the formulas given in previous studies [34].

Images of yeast and mammalian cells were recorded in areas of 10 µm × 10 µm and 30 µm × 30 µm with a resolution level of 100 nm and 250 nm, respectively. Small images of yeast cells were obtained in areas of 1 µm × 1 μm of the cell surface, with a resolution level of 30 nm. For each concentration point, three to five large images were obtained, for a total of 5 to 10 cells. Image processing was performed using the “SICMImageViewer” software (version 1.2.5.10, ICAPPIC Ltd, Lonon, UK). Antifungal efficacy analysis was provided by comparing the heights of the bulging areas of a cell wall based on the surface profiles.

### 2.5. Amperometric ROS Measurements in a Single Cell

The total ROS concentration was determined by the amperometric method using Pt-nanoelectrodes. Commercially available disk-shaped carbon nanoelectrodes isolated in quartz (ICAPPIC Limited, UK), with diameters of 60–100 nm were used to prepare Pt nanoelectrodes. Before the deposition of platinum on the carbon surface, the disk-shaped carbon electrode was etched in a 0.1 M NaOH and 10 mM KCl solution for 40 cycles of 10 seconds (from 0 to +2200 mV) to create nanocavities. Further electrochemical deposition of platinum in nanocavities was achieved by cycling from 0 to 800 mV, with a scan rate of 200 mV/s for 4 to 5 cycles in a 2 mM H2PtCl6 solution in 0.1 M hydrochloric acid. Cyclic voltammetry from −800 to 800 mV, with a scan rate of 400 mV/s, in a 1 mM solution of ferrocenmethanol in PBS was used to control the electrode surface at all stages of fabrication. Prior to the measurements, each platinum nanoelectrode was calibrated using a series of standard H_2_O_2_ solutions at a potential of +800 mV. The preparation of Pt-nanoelectrodes has been described in detail elsewhere [40,41].

PC3 (2 × 10^5^) and HEK293 (3 × 10^5^) cells were seeded in 35 mm Petri dishes and treated the next day with compounds. The compounds were dissolved in DMSO and diluted in culture medium. The final concentration of the compounds in the culture medium corresponded to MIC (minimum inhibitory concentration) values after 1 h of incubation. MIC values were determined using the standard broth microdilution method. Untreated cells were used as a control, which was performed at the beginning and end of the experiment. After the incubation time, attached cells in Petri dishes were washed three times with Hanks’ Balanced Salt Solution to remove the growth medium and traces of compounds. A nanoelectrode penetrated the cells and measured the oxidation current of hydrogen peroxide. On average, approximately 15 cells were measured by 2–4 Pt electrodes in three independent Petri dishes for each compound.

The potential difference between the Pt nanoelectrode and the reference electrode was recorded by a patch-clamp amplifier MultiClamp 700B (Axon Instruments, Burlingame, CA, USA) and transferred to a computer using the ADC-DAC converter Axon Digidata 1550B (Axon Instruments, Burlingame, CA, USA) and the pClamp 11 software suite (pClamp 11.0.0.3, Molecular Devices, Silicon Valley, CA, USA). A micromanipulator PatchStar (Scientifica, Uckfield, UK) was used to manipulate the nanoelectrode. All manipulations were made on an optical inverted microscope Eclipse Ti-U (Nikon, Tokyo, Japan) located on a vibration isolation table (Supertech Instruments, London, UK). The current inside the cell was measured, and the obtained data was processed using the Origin 2021 software (version 9.8, Origin Lab, Northampton, MA, USA). The total intracellular ROS level was determined based on the calibration curve.

## 3. Results and Discussion

### 3.1. Effect of the Drug L-173 on Candida *spp.*

In the control group of *C. parapsilosis* ATCC 22019, the cell shape is oval or round, the surface is smooth, and the size varies from 2 to 15 µm (Figure 1A). Cells of the control group *C. albicans* 8R are rounded (from 2 to 7 μm in size), their surface is smooth, and the relief is poorly developed (Figure 1B). Cells of the control group of *C. krusei* 432M have the shape of an elongated ellipse (size from 3 to 10 μm), their surface is rough, and they form agglomerates up to 60 μm in diameter (Figure 1C).

Based on the obtained topography data, the following types of damage were detected: the formation of a tear-off crack on the cell surface (example in Figure 1D, inset); the formation of flocculent structures on the cell surface, which we interpret as destruction of the cell wall (example in Figure 1E, inset); and the formation of fibrous structures on the cell surface, which we interpret as a destroyed polysaccharide capsule (example in Figure 1F, upper right corner).

During the incubation of *C. parapsilosis* ATCC 22019 with the substance L-173 (Figure 1S) at a concentration of c = 40 µg/mL (as well as 10 µg/mL and 20 µg/mL) during the day, destruction of the cell wall of the fungus was revealed, the surface of which became loose (Figure 1D). The number of damaged areas is relatively small, and their average height is about 0.2 μm, which may be due to the species-specific features of this strain. When inhibited at low concentrations (1 μg/mL and 5 μg/mL), tear-off cracks formed on the cell surface. *C. albicans* 8R cells incubated with L-173 (c = 40 μg/mL) had a wide zone of damage to the cell wall and surface polysaccharides, the height of which reached 0.9 μm (Figure 1E). At low concentrations of L-173 (1 μg/mL and 5 μg/mL), no effect was found, and at concentrations of 10 μg/mL and 20 μg/mL, damaged areas of the cell wall were found on the surface. On the surface cells of *C. krusei*, treated with the drug L-173 (c = 40 µg/mL), destroyed structures of polysaccharides (up to 1.5 µm high) and damaged areas of the cell wall (up to 0.5 µm high) were found (Figure 1F). At a low concentration of 1 μg/mL, no effect was found; at a concentration of 5 μg/mL, slight changes in the surface structure were found; and at concentrations of 10 μg/mL and 20 μg/mL, destruction of the cell wall was observed on the cell surface.

To compare the effects of the drug L-173 at different concentrations, the following classification was used: “0”: the effect of the drug was not detected, the cells are similar to the control group; “1”: tear-off cracks were found on the cell surface; “2”: the cell wall is destroyed on its surface, flaky and fibrous structures are formed; “3”: the surface of the cells is damaged by more than 80%; the cells are considered dead/destroyed. The data are presented in Table 1.

### 3.2. Effects of Antimicrobials on Mammalian Cells

In addition to the above work, the effect of the antimicrobials fluconazole and L-173 on human prostate cancer cell lines PC3 and human embryonic kidney HEK 293 was studied. Topography (A1–F1) and a map of mechanical properties (A2–F2) are presented in Figure 2.

According to the data obtained, it was found that when exposed to fluconazole or L-173 on PC3 cancer cells, cell stiffness increases. For control cells, cells with the addition of fluconazole, and cells with the addition of drug L-173, Young’s modulus, E = (0.80 ± 0.01) kPa; E = (1.02 ± 0.02) kPa; and E = (1.37 ± 0.05) kPa respectively. In the case of treatment of healthy human cells with HEK 293 antifungal drugs, no significant difference was found between the stiffness of control cells, cells treated with fluconazole, and cells treated with the drug L-173, and their Young’s modulus is E = (0.77 ± 0.01) kPa; E = (0.78 ± 0.01) kPa; E = (0.79 ± 0.01) kPa, respectively. Figure 3 shows a histogram of the obtained Young’s modulus for PC3 and HEK 293 cells. The observed stiffening of the cytoskeleton of the PC3 cell line indicates the cytostatic effect of the drug of the azole group fluconazole, and the substance L-173, of the thiazolidinedione group. This is consistent with the previously observed cytostatic effects of the drugs in these groups [42,43,44,45]. In addition, the absence of changes in the mechanical properties of a healthy HEK 293 cell line when exposed to fluconazole and L-173 may indicate a low toxicity of these drugs.

### 3.3. Intracellular ROS Level Measurement

Amperometric quantitative detection of the total ROS level using Pt-nanoelectrodes is a stable and well-established method for biological applications. Such low-invasive nanoelectrodes are highly sensitive and allow real-time measurements without the preliminary preparation of the studied samples [36,37,40,41,46,47]. In this case, the studies were carried out to measure ROS level inside prostate cancer cells (PC3) and human embryonic kidney cells (HEK 293). For the HEK 293 cell line, the following values of the intracellular ROS level were obtained under the influence of the compounds (Figure 3). At least 10 cells were measured using a Pt-nanoelectrode; it was shown that fluconazole and L-173 influenced cells and the generation of ROS followed. In this case, a significant increase (up to 2×) in ROS level was observed under the influence of fluconazole. The synthesized substance (L-173) did not exhibit statistically significant activity against HEK 293 and is less toxic to this cell line than fluconazole. In the course of the experiment in relation to the PC3 cell line, fluconazole exposure led to an increase in ROS expression compared to the control; however, the difference was not statistically significant. Meanwhile, exposure to L-173 led to an almost threefold increase in ROS expression in PC3 cells (Figure 3). The data obtained are consistent with the concept of anticancer activity of the preparations of the azole and thiazolidinedione groups [48,49].

### 3.4. Cytotoxicity Assay

The obtained compounds were tested for cytotoxicity using the standard MTS assay. This study was realized using the non-cancer human embryonic kidney cell line HEK293 and the human prostate cancer cell line (PC3). The reasons for the choice of cell lines were that, firstly, these cell lines are the most widely used in cell studies and, secondly, there is research in which the effect of compounds with an azole group on these cell lines has previously been investigated [50]. The results of this assay are shown in Figure 3B. The MTS assay revealed that fluconazole did not cause cytotoxic effects at any tested concentrations. In contrast, LEV-173 resulted in cell death activation (Figure 3B): the IC_50_ for the PC-3 cell line was 34.9 ± 0.26 uM; for HEK293 culture, it was 16.6 ± 0.2 uM.

## 4. Conclusions

The wide range of applications of the SICM method and the ICM application for measuring ROS make it possible to answer questions in various research directions. Thus, in this work, the effect of the drug L-173 as an antifungal (to pathogenic species in the genus *Candida*) and anticancer (on a human prostate cancer cell line) agent was studied. We obtained images of the control groups of the yeast strains considered in this work, all of them showing smooth surface profiles without structural anomalies. Then, the test samples were incubated with the studied drugs. The results obtained demonstrate that daily incubation with L-173 in *C. parapsilosis* ATCC 22019 cells causes damage to the cell wall at high and medium concentrations of the drug. At low concentrations, activity is also observed, but it is weakly expressed. In general, L-173 has a stable and moderate effect on *C. parapsilosis* ATCC 22019 cells. A different picture is observed when *C. albicans* are exposed to 8R L-173. For this species, at low concentrations of the drug, the effect is not manifested; however, starting from a concentration of 10 μg/mL, the structures of the cell wall and polysaccharides are extensively destroyed. As for the strain *C. krusei* 432M, no effect is observed at a low concentration of the drug, but with an increase in the dose, a gradual degradation of surface structures occurs, demonstrated by, not only the destruction of the cell wall, but also the decay of polysaccharides. A comparative evaluation of the action of the preparation L-173 on yeast showed that the greatest antifungal effect was observed in *Candida albicans*. With this species, L-173 was more stable at all doses considered; however, L-173 was less active against *Candida parapsilosis*, and it had average activity at medium and high doses against the species *Candida krusei*.

It was determined that the antifungal drugs fluconazole and L-173 exhibit a pronounced cytostatic effect on PC3 cancer cells. The Young’s modulus of the cells treated with fluconazole increased approximately one and a half times, and in the cells treated with L-173, rigidity doubled. Moreover, the drugs do not act on healthy HEK 293 cells, which may indicate their low toxicity to mammalian cells. Specifically, the Young’s modulus of HEK 293 cells treated with fluconazole or L-173 were statistically indistinguishable from the elastic modulus of control cells. It was found that fluconazole causes a twofold increase in ROS in HEK cells, while the newly synthesized substance L-173 does not lead to the expression of ROS. Substance L-173 also exhibits anticancer activity by expressing ROS in PC3 cancer cells, which can also lead to their apoptosis. It should be noted that the new compound is promising in terms of safety, as it affects healthy HEK 293 cells to a lesser extent than does the comparator.

The presented work demonstrates the expansion of the application of ICM methods in the areas of biophysics, cell biology, and pharmacology. We demonstrate a new approach in the study of the antimicrobial and anticancer efficacy of new-generation drugs. The innovative ICM method provides information about the topography, mechanical, and biophysical properties of biological objects. Further, an integrated ICM approach can be successfully applied in solving the issues of preclinical trials of new antifungal drugs or the fundamental study of the surface structures and their mechanical properties of yeast.

## Figures and Tables

**Figure 1 cells-12-01666-f001:**
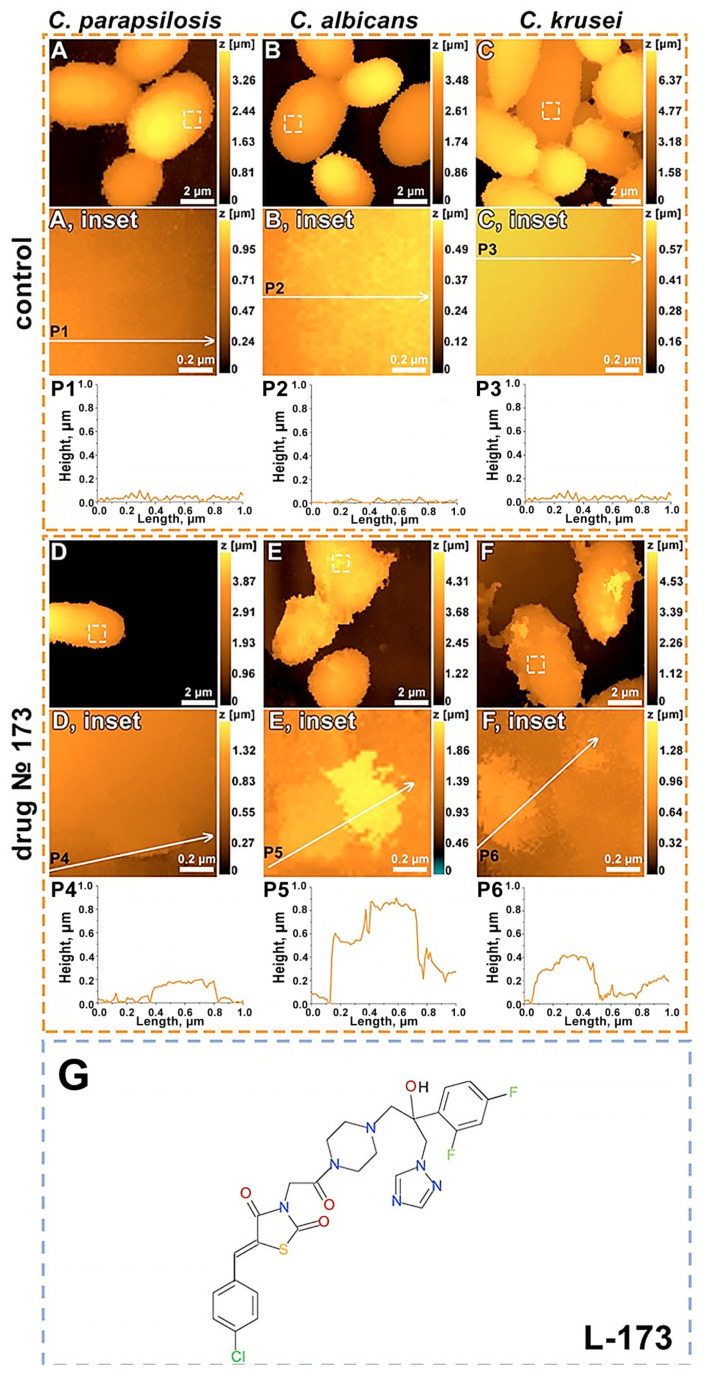
SICM images of *C. parapsilosis* ATCC 22019 (**A**,**D**), *C. albicans* 8R (**B**,**E**), *C. krusei* 432M (**C**,**F**). Control cells (**A**–**C**) and cells after L-173 treatment at 40 μg/mL (**D**–**F**). White dashed line squares indicate the enlarged areas on the insets (**A**–**E**). White lines mark profiles (**P1**–**P6**). The structure of the 1,2,4-triazole derivative L-173 (**G**).

**Figure 2 cells-12-01666-f002:**
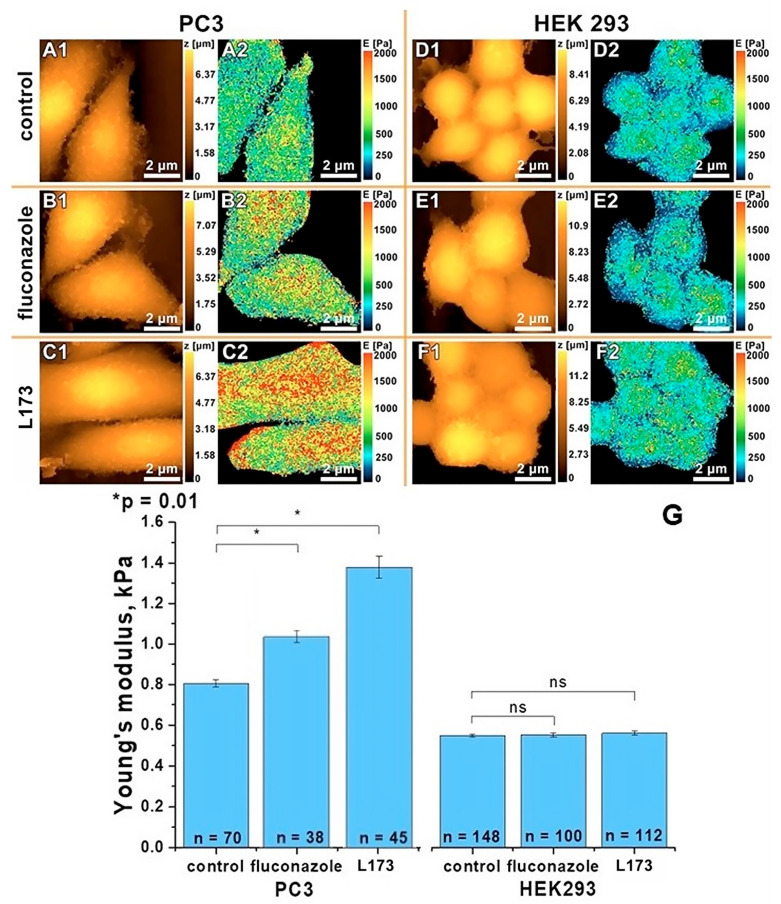
SICM topography (**1**) and a Young’s modulus map (**2**) of control cells (**A**,**D**); cells treated with fluconazole (**B**,**E**); cells treated with drug L-173 (**C**,**F**), where PC3 is in (**A**–**C**) and HEK 293 in (**D**–**F**). Histogram of PC3 and HEK 293 cell stiffness with the addition of antifungals, SE and (*) *p* < 0.01 (one-way ANOVA) (“ns” is not significant) (**G**).

**Figure 3 cells-12-01666-f003:**
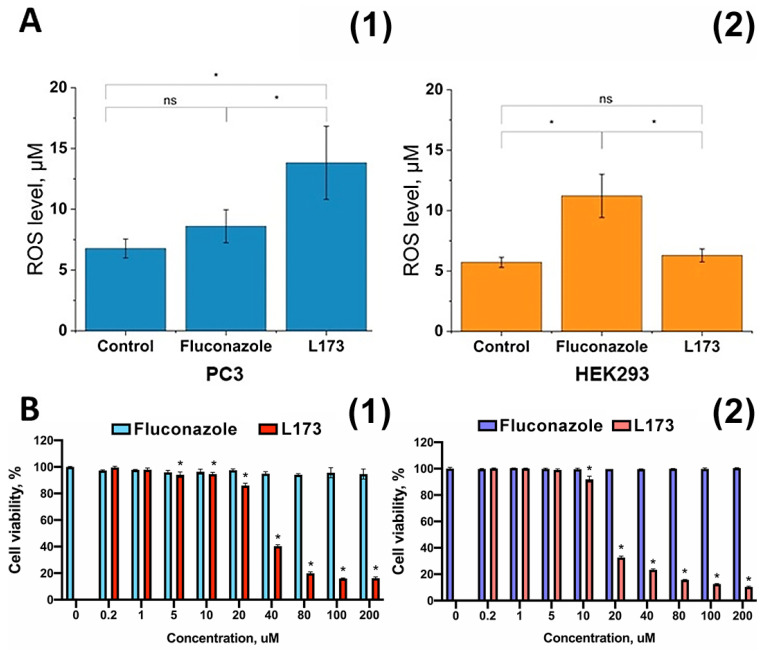
(**A**) The level of ROS inside PC3 cells (**1**) and HEK293 (**2**) under the influence of drugs. The results are shown as mean, SE and (*) *p* < 0.05 (one-way ANOVA). (**B**) Cell viability assessment after 72 h of incubation with various concentrations of fluconazole or L173 by MTS test. Results are shown as means ± SD, * *p* < 0.05 (*t*-test) comparing cells incubated in culture medium (“ns” is not significant). (**1**) PC-3 cell line; (**2**) HEK293 culture.

**Table 1 cells-12-01666-t001:** Effect of the drug L-173 on *Candida* yeast (where 0 is absence of effect and 3 is maximum effect).

*Candida* Species	Drug Concentration L-173, µg/mL				
	1	5	10	20	40
*C. parapsilosis* ATCC 22019	1	1	2	2	2
*C. albicans* 8R	0	0	2	2	3
*C. krusei* 432M	0	1	2	2	3

## Data Availability

The data used in the publication is not hosted on any resources and can be provided privately upon request.

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
