# Peer review of "Investigation of the Antifungal and Anticancer Effects of the Novel Synthesized Thiazolidinedione by Ion-Conductance Microscopy"

_cells, 2023, doi:10.3390/cells12121666_

Round 1
Reviewer 1 Report
Savin et al. present a study of the novel synthesized thiazolidinedione regarding its antimicrobial, anticancer properties as well as its toxicity which was conducted by using ion- conductance microscopy and an amperometric method. The authors do a nice job presenting the background and why it is essential to develop and study new antifungal drugs, stating clearly the objective of their work and its significance. They are also describing the materials and methods used while presenting their findings and conclusions. This reviewer is positive as the work is interesting but there are some points that need to be altered as written in more details below.
Page 2, line 85: “Scanning ion-conductance microscopy is a multifunctional technologyand a very 85 promising method”. A space is needed between “technology” & “and”.
Page 3, Line 110-111: “Then the cups was washed with distilled water, dried, and a 2% glutaraldehyde solution is applied”. There are some typos here. “The cup was (or the cups were?) then washed with distilled water, dried and finally a 2% glutaraldehyde solution was applied”. The authors should use one form of tense to describe the methods. It is recommended that since the authors use past tense most of the times, they continue using past tense everywhere.
Page 3, Lines 117-121: “Weighed samples of the was dissolved in a 99.9% dimethyl sulfoxide to a concentration of 0.1 mg/ml, and then brought to the test concentrations in 1 ml of Hank's solution. The washed of dishes were filled with 2 ml of Hank's solution and placed on the scanning platform. The samples was incubate with drug during the day”
There are many typos in general in this paragraph. The above sentences have many of them.
Weighed samples of the .. ? what? A noun is missing here and it’s hard for the reader to understand the meaning of the sentence. Then the authors write the washed of dishes”
Please make sure that you are using the current form of each verb when referring to singular or plural. For examples the authors write: “the samples was incubate” This sentence is not grammatically correct. It should be “the samples were placed into an incubator with drug”? or “the samples were incubated?” Please change the sentences and make sure there are no such typos/mistakes in the manuscript so that the reader can fully understand the work that has been done.
Page 4, Line 182: “The compounds was dissolve in DMSO and was dilute in culture medium” Again same typos here. It should be “the compounds were dissolved in DMSO and then were diluted in..”
Page 5, Table 1: Please include more details in the caption of the table so that the reader can fully understand what the table shows by reading its caption. It is highly recommended that the authors include the meaning of the numbers “0, 1, 2, 3” in the caption as well in addition to the description that they write in lines 220-224.
Page 5, Line 223-224: “dead/destroyed cells 223 was founds”. There is another typo here it should be “were found”. Please make sure that the manuscript is corrected.
Page 6, Figure 1: Overall image quality is poor. Especially when the figure is zoomed in, everything becomes faded and blurry. The image quality should be good when zoomed in so that the reader can easily look at the figure and understand its meaning. Also, it appears that the scale bars of A, B, C, D, E and F insets are written incorrectly. There should be a dot instead of a comma in “0,2 μm”. That should be written as “0.2 μm”. Also, the title of the y axis in all figures P1-P6 has a typo, it should be “Length” instead of “Lenght”. Also the numbers in both x and y axes are written incorrectly as well, there should be a dot instead of a comma (e.g. 0.6 instead of 0,6) It is highly recommended that the authors make sure that all the numbers are written in a correct format in this manuscript. Finally, it would be easier for the reader if the authors increase the size of the numbers in all the plots so that the reader doesn’t have to focus a lot. Finally in the caption of Figure 1 it is written: ”White lines mark profiles (1P – 6P)”. Since in the figure the authors write P1 and P6 (and not 1P and 6P), the same wording should be followed in the caption as well so that the reader will be sure to what this refers to. Also, please include a more detailed description in the figure caption of all the subfigures of Figure 1 so that the figure is more complete with its caption.
Page 7, Line 266: “Obtained compounds was teste for cytotoxicity”. There is a typo here it should be “obtained compounds were tested for cytotoxicity”.
Page 7, Line 278: “the effect of drug L-173 as an antifungal (to Candida specie) and anticancer (on a human 278 prostate cancer cell line) agent are studied”. There is another typo here it should be “the effect … was studied”.
Page 8, Line 283-284: “A comparative evaluation of the action of drug L-173 on yeast showed 283 that the greatest antifungal effect are observe in relation to Candida albicans specie.”
There are typos here it should be: “A comparative evaluation… effect was observed in relation to Candida albicans species”
Figures 2 & 3: The same corrections as written previously for Figure 1 apply here for both figures as well. For example, the numbers should be written with a dot instead of a comma. Also poor image quality in both figures that should be addressed, while captions need to be written with more details when describing a figure.
Conclusions section: The conclusions are written in a good format but it is recommended that the authors elaborate a little bit more on the findings and the conclusions here in a clear way so that the conclusion part of the manuscript will be stronger and more complete.
There are several typos and phrases that are incorrect in English that need to be corrected. More details have been provided in the Comments and Suggestions for Authors section with specific examples.
Author Response
Dear reviewer,
We would like to thank you for taking the necessary time and effort to review the manuscript. We sincerely appreciate all your valuable comments and suggestions, which helped us in improving the quality of the manuscript.
Major paper changes.
- The manuscript has passed a grammatical check.
- Descriptions of "Results and Methods" have been changed.
- Image quality has been improved.
- Expanded block "Conclusions".
- Rewritten block "3.1".
- Figure captions have been added.
Point 1: Page 2, line 85: “Scanning ion-conductance microscopy is a multifunctional technology and a very 85 promising method”. A space is needed between “technology” & “and”.
Response 1: Corrected.
Point 2: Page 3, Line 110-111: “Then the cups was washed with distilled water, dried, and a 2% glutaraldehyde solution is applied”. There are some typos here. “The cup was (or the cups were?) then washed with distilled water, dried and finally a 2% glutaraldehyde solution was applied”. The authors should use one form of tense to describe the methods. It is recommended that since the authors use past tense most of the times, they continue using past tense everywhere.
Response 2: Corrected.
Point 3: Page 3, Lines 117-121: “Weighed samples of the was dissolved in a 99.9% dimethyl sulfoxide to a concentration of 0.1 mg/ml, and then brought to the test concentrations in 1 ml of Hank's solution. The washed of dishes were filled with 2 ml of Hank's solution and placed on the scanning platform. The samples was incubate with drug during the day”
There are many typos in general in this paragraph. The above sentences have many of them.
Weighed samples of the .. ? what? A noun is missing here and it’s hard for the reader to understand the meaning of the sentence. Then the authors write the washed of dishes”
Please make sure that you are using the current form of each verb when referring to singular or plural. For examples the authors write: “the samples was incubate” This sentence is not grammatically correct. It should be “the samples were placed into an incubator with drug”? or “the samples were incubated?” Please change the sentences and make sure there are no such typos/mistakes in the manuscript so that the reader can fully understand the work that has been done.
Response 3: Corrected.
Point 4: Page 4, Line 182: “The compounds was dissolve in DMSO and was dilute in culture medium” Again same typos here. It should be “the compounds were dissolved in DMSO and then were diluted in..”
Response 4: Corrected.
Point 5: Page 5, Table 1: Please include more details in the caption of the table so that the reader can fully understand what the table shows by reading its caption. It is highly recommended that the authors include the meaning of the numbers “0, 1, 2, 3” in the caption as well in addition to the description that they write in lines 220-224.
Response 5: Corrected.
Point 6: Page 5, Line 223-224: “dead/destroyed cells 223 was founds”. There is another typo here it should be “were found”. Please make sure that the manuscript is corrected.
Response 6: Corrected.
Point 7: Page 6, Figure 1: Overall image quality is poor. Especially when the figure is zoomed in, everything becomes faded and blurry. The image quality should be good when zoomed in so that the reader can easily look at the figure and understand its meaning. Also, it appears that the scale bars of A, B, C, D, E and F insets are written incorrectly. There should be a dot instead of a comma in “0,2 μm”. That should be written as “0.2 μm”. Also, the title of the y axis in all figures P1-P6 has a typo, it should be “Length” instead of “Lenght”. Also the numbers in both x and y axes are written incorrectly as well, there should be a dot instead of a comma (e.g. 0.6 instead of 0,6) It is highly recommended that the authors make sure that all the numbers are written in a correct format in this manuscript. Finally, it would be easier for the reader if the authors increase the size of the numbers in all the plots so that the reader doesn’t have to focus a lot. Finally in the caption of Figure 1 it is written: ”White lines mark profiles (1P – 6P)”. Since in the figure the authors write P1 and P6 (and not 1P and 6P), the same wording should be followed in the caption as well so that the reader will be sure to what this refers to. Also, please include a more detailed description in the figure caption of all the subfigures of Figure 1 so that the figure is more complete with its caption.
Response 7: Unfortunately, it is not possible to improve the image quality of zoomed in figures since this is the maximum instrumental resolution. Each pixel represents a probe approach point to the surface. And the distance between two such points is equal to the diameter of the nanocapillary (in our case, the minimum possible diameter is 40 nm). Other comments have been corrected.
Point 8: Page 7, Line 266: “Obtained compounds was teste for cytotoxicity”. There is a typo here it should be “obtained compounds were tested for cytotoxicity”.
Response 8: Corrected.
Point 9: Page 7, Line 278: “the effect of drug L-173 as an antifungal (to Candida specie) and anticancer (on a human 278 prostate cancer cell line) agent are studied”. There is another typo here it should be “the effect … was studied”.
Response 9: Corrected.
Point 10: Page 8, Line 283-284: “A comparative evaluation of the action of drug L-173 on yeast showed 283 that the greatest antifungal effect are observe in relation to Candida albicans specie.”
There are typos here it should be: “A comparative evaluation… effect was observed in relation to Candida albicans species”
Response 10: Corrected.
Point 11: Figures 2 & 3: The same corrections as written previously for Figure 1 apply here for both figures as well. For example, the numbers should be written with a dot instead of a comma. Also poor image quality in both figures that should be addressed, while captions need to be written with more details when describing a figure.
Response 11: Corrected.
Point 12: Conclusions section: The conclusions are written in a good format but it is recommended that the authors elaborate a little bit more on the findings and the conclusions here in a clear way so that the conclusion part of the manuscript will be stronger and more complete.
Response 12: Corrected.

Reviewer 2 Report
The authors submited a manuscript reporting the antifungal and anticancer activity of a novel thiazolidinedione (L-173).
Overall, the manuscript is well writen but a few things need to be cleared:
Line 18 - it is HEK293 cells not HEK239
Line 68 - Candida glabrata shouldn't be in italic?
Lines 114, 127, 128, 179, 181 -superscript and subscript should be
Line 118 - "Weighed samples of 117 the was dissolved in a 99.9% dimethyl sulfoxide to a concentration of 0.1 mg/ml". It is not clear was the 99.9% is referred to. Is it the purity of DMSO? This should be clarified.
Line 182 - "dissolve" should be "dissolved"
Line 266 - "was teste" should be "was tested"
Line 272/Figure 3 - it is not clear the difference between IC50 and CC50. The CC50 is nowhere described in the manuscript.
minor english editing is required.
Author Response
Dear reviewer,
We would like to thank you for taking the necessary time and effort to review the manuscript. We sincerely appreciate all your valuable comments and suggestions, which helped us in improving the quality of the manuscript.
Major paper changes.
- The manuscript has passed a grammatical check.
- Descriptions of "Results and Methods" have been changed.
- Image quality has been improved.
- Expanded block "Conclusions".
- Rewritten block "3.1".
- Figure captions have been added.
Point 1: Line 18 - it is HEK293 cells not HEK239
Response 1: Corrected.
Point 2: Line 68 - Candida glabrata shouldn't be in italic?
Response 2: Corrected.
Point 3: Lines 114, 127, 128, 179, 181 -superscript and subscript should be/
Response 3: Corrected.
Point 4: Line 118 - "Weighed samples of 117 the was dissolved in a 99.9% dimethyl sulfoxide to a concentration of 0.1 mg/ml". It is not clear was the 99.9% is referred to. Is it the purity of DMSO? This should be clarified.
Response 4: Corrected.
Point 5: Line 182 - "dissolve" should be «dissolved».
Response 5: Corrected.
Point 6: Line 266 - "was teste" should be "was tested".
Response 6: Corrected.
Point 7: Line 272/Figure 3 - it is not clear the difference between IC50 and CC50. The CC50 is nowhere described in the manuscript.
Response 7: We have carried out the MTS test again. In the latest version of the publication, CC50 is not mentioned as unnecessary.

Reviewer 3 Report
The present research study entitled "Investigation of the antifungal and anticancer effects of the novel synthesized thiazolidinedione by ion- conductance microscopy" investigates the effect of L-173 as a thiazolidinedione and its antimicrobial (for Candida spp.), anticancer properties (on samples of the human prostate cell line PC3) and drug toxicity (on a sample of the human kidney cell line HEK239) using scanning ion-conductance microscopy (SICM) technique. The experimental results revealed that the proposed compound exert antifungal and anticancer activity by disrupting the cell walls and ROS generation. The results are interesting, however, the manuscript needs major modifications and some issues should be addressed as follows:
- The quality of English language should be increased. There are numerous typos and grammatical errors needed to be refined thoroughly.
- In page 2, "SIPM" should be changed to "SICM".
- The figure captions are not very clear. Please rewrite the captions to be more understandable.
- In Figure 1D, the SICM image of only one specie is presented. How many species have been studied? Was the same effect observed in other samples as well.
- The process of electrode preparation and the experimental set-up is better to be presented for clarification.
- What is the advantage of the method presented here for measuring ROS generation in comparison to other methods such as DCFH-DA analysis?
- The quality of English language should be increased. There are numerous typos and grammatical errors needed to be refined thoroughly.
Author Response
Dear reviewer,
We would like to thank you for taking the necessary time and effort to review the manuscript. We sincerely appreciate all your valuable comments and suggestions, which helped us in improving the quality of the manuscript.
Major paper changes.
- The manuscript has passed a grammatical check.
- Descriptions of "Results and Methods" have been changed.
- Image quality has been improved.
- Expanded block "Conclusions".
- Rewritten block "3.1".
- Figure captions have been added.
Point 1: The quality of English language should be increased. There are numerous typos and grammatical errors needed to be refined thoroughly.
Response 1: Corrected.
Point 2: In page 2, "SIPM" should be changed to "SICM".
Response 2: Corrected.
Point 3: The figure captions are not very clear. Please rewrite the captions to be more understandable.
Response 3: Figure captions have been rewritten.
Point 4: In Figure 1D, the SICM image of only one specie is presented. How many species have been studied? Was the same effect observed in other samples as well.
Response 4: The manuscript presents representative results for each species. For each concentration point, three to five large images were obtained, for a total of 5 to 10 cells.
Point 5: The process of electrode preparation and the experimental set-up is better to be presented for clarification.
Response 5: It was included in Section 2.4.
Point 6: What is the advantage of the method presented here for measuring ROS generation in comparison to other methods such as DCFH-DA analysis?
Response 6: Compared to fluorescent methods (in particular, the DCFH-DA assay), ROS detection using Pt-nanoelectrode allows quantitative detection with higher sensitivity, which is an extremely important task during the analysis of single cells with a low total ROS level. This conclusion was the result of one of our early research projects in which we compared DCFH-DA and a Pt-nanoelectrode (Erofeev et al., 2018). Subsequently, we have used this method in other research containing in vitro and in vivo measurements (Krasnovskaya et al., 2020; A. E. Machulkin et al., 2021; Novotortsev et al., 2021; Petrov et al., 2021b; Vaneev et al., 2020). In addition, we added a small introduction to Section 3.3., where we emphasized the feasibility of using nanoelectrodes.
- Vaneev, A.N.; Gorelkin, P. V.; Garanina, A.S.; Lopatukhina, H. V.; Vodopyanov, S.S.; Alova, A. V.; Ryabaya, O.O.; Akasov, R.A.; Zhang, Y.; Novak, P.; et al. In Vitro and In Vivo Electrochemical Measurement of Reactive Oxygen Species After Treatment with Anticancer Drugs. Anal Chem 2020, 92, 8010–8014, doi:10.1021/acs.analchem.0c01256.
- Machulkin, A.; Uspenskaya, A.; Zyk, N.; Nimenko, E.; Ber, A.; Petrov, S.; Shafikov, R.; Skvortsov, D.; Smirnova, G.; Borisova, Y.; et al. PSMA-Targeted Small-Molecule Docetaxel Conjugate: Synthesis and Preclinical Evaluation. Eur J Med Chem 2021, 227, 113936, doi:10.1016/j.ejmech.2021.113936.
- Krasnovskaya, O.O.; Guk, D.A.; Naumov, A.E.; Nikitina, V.N.; Semkina, A.S.; Vlasova, K.Yu.; Pokrovsky, V.; Ryabaya, O.O.; Karshieva, S.S.; Skvortsov, D.A.; et al. Novel Copper-Containing Cytotoxic Agents Based on 2-Thioxoimidazolones. J Med Chem 2020, 63, 13031–13063, doi:10.1021/acs.jmedchem.0c01196.
- Petrov, R.A.; Mefedova, S.R.; Yamansarov, E.Yu.; Maklakova, S.Yu.; Grishin, D.A.; Lopatukhina, E. V.; Burenina, O.Y.; Lopukhov, A. V.; Kovalev, S. V.; Timchenko, Y. V.; et al. New Small-Molecule Glycoconjugates of Docetaxel and GalNAc for Targeted Delivery to Hepatocellular Carcinoma. Mol Pharm 2021, 18, 461–468, doi:10.1021/acs.molpharmaceut.0c00980.
- Erofeev, A.; Gorelkin, P.; Garanina, A.; Alova, A.; Efremova, M.; Vorobyeva, N. Novel Method for Rapid Toxicity Screening of Magnetic Nanoparticles. Sci Rep 2018, 1–11, doi:10.1038/s41598-018-25852-4.
- Novotortsev, V.K.; Kukushkin, M.E.; Tafeenko, V.A.; Skvortsov, D.A.; Kalinina, M.A.; Timoshenko, R. V.; Chmelyuk, N.S.; Vasilyeva, L.A.; Tarasevich, B.N.; Gorelkin, P. V.; et al. Dispirooxindoles Based on 2-Selenoxo-Imidazolidin-4-Ones: Synthesis, Cytotoxicity and ROS Generation Ability. Int J Mol Sci 2021, 22, 2613, doi:10.3390/ijms22052613.

Reviewer 4 Report
In this manuscript, the anticancer and antifungal effects of the newly synthesized thiazolidinedione (L-173) were studied. Scanning ion-conductance microscopy was used to detect the morphology of candida and the modulus of cancer cells after drug treatment. Platinum nanoelectrode was used to detect the effect of drugs on intracellular ROS levels. The toxicity and inhibitory effect of the drug on normal and cancer cells were measured by standard MTS method.
1. When exploring the influence of L-173 on the morphology of Candida, it was described in the paper that L-173 caused surface damage of C. krusei of 1.5 µm, while the cell surface damage shown in Fig. 1F was less than 0.5 µm, which was inconsistent with the description in the paper. When further exploring the influence of L-173 concentration, the numbers "0", "1", "2", "3" in Table 1 can directly reflect the gradual enhancement of drug effect, but it does not mention how to distinguish between "1" and "2" which is corresponding to the changes of cell wall and the protrusion of cell membrane. If differentiated by modulus differences, showing the surface mechanical properties of Candida will be helpful to illustrate drug effects.
2. In Fig. 3B, CC50 refers to the drug concentration corresponding to 50% cell death. This determination process is generally to set up gradient dilution of drug concentration and draw a dose-dependent regression curve of cell activity to determine the corresponding value of CC50. The standard error of the CC50 for Fluconazole in the tables for HEK293 and PC3 is large, probably because the concentration gradient is set up too widely. Adding a bar chart showing the cell activity is helpful to illustrate the effect of drug concentration on cell activity, and the establishment of more concentration gradients is helpful to the accurate quantification of CC50 and IC50.
3. The clarity of figures in the paper is poor, and some numbers are on the upper left and some on the upper right. It is suggested to unify their positions; Moreover, the same noun font is not uniform in many places in the text. For example, ‘parapsilosis’ in Line 104 is italic, while in Line 210 it is positive.
4. Figures are used in Table 1 to distinguish the effects of drugs of different concentrations on different strains, but what are the criteria for these levels of differentiation? If the differentiation is based on the height change of cell surface structure before and after drug action, is specific threshold set for differentiation?
5. Fig. 3B is presented in the form of a table, and the expression of the results is not intuitive enough. It is suggested to present the results with figures and supplemented with explanations.
Minor editing of English language required.
Author Response
Dear reviewer,
We would like to thank you for taking the necessary time and effort to review the manuscript. We sincerely appreciate all your valuable comments and suggestions, which helped us in improving the quality of the manuscript.
Major paper changes.
- The manuscript has passed a grammatical check.
- Descriptions of "Results and Methods" have been changed.
- Image quality has been improved.
- Expanded block "Conclusions".
- Rewritten block "3.1".
- Figure captions have been added.
Point 1: When exploring the influence of L-173 on the morphology of Candida, it was described in the paper that L-173 caused surface damage of C. krusei of 1.5 µm, while the cell surface damage shown in Fig. 1F was less than 0.5 µm, which was inconsistent with the description in the paper. When further exploring the influence of L-173 concentration, the numbers "0", "1", "2", "3" in Table 1 can directly reflect the gradual enhancement of drug effect, but it does not mention how to distinguish between "1" and "2" which is corresponding to the changes of cell wall and the protrusion of cell membrane. If differentiated by modulus differences, showing the surface mechanical properties of Candida will be helpful to illustrate drug effects.
Response 1: Figure 1F shows the destruction of the cell wall, which corresponds to a height of up to 0.5 µm. As indicated in the text, the value of 1.5 µm corresponds to the polysaccharides formed on the surface. We made an inaccuracy, "2" means the formation of bulging structures on the surface, not the cell membrane.
Clarifications on this issue have been made in the text of the article.
Point 2: In Fig. 3B, CC50 refers to the drug concentration corresponding to 50% cell death. This determination process is generally to set up gradient dilution of drug concentration and draw a dose-dependent regression curve of cell activity to determine the corresponding value of CC50. The standard error of the CC50 for Fluconazole in the tables for HEK293 and PC3 is large, probably because the concentration gradient is set up too widely. Adding a bar chart showing the cell activity is helpful to illustrate the effect of drug concentration on cell activity, and the establishment of more concentration gradients is helpful to the accurate quantification of CC50 and IC50.
Response 2: We repeated the MTS test with more accuracy.
Point 3: The clarity of figures in the paper is poor, and some numbers are on the upper left and some on the upper right. It is suggested to unify their positions; Moreover, the same noun font is not uniform in many places in the text. For example, ‘parapsilosis’ in Line 104 is italic, while in Line 210 it is positive.
Response 3: Corrected.
Point 4: Figures are used in Table 1 to distinguish the effects of drugs of different concentrations on different strains, but what are the criteria for these levels of differentiation? If the differentiation is based on the height change of cell surface structure before and after drug action, is specific threshold set for differentiation?
Response 4: Differentiation according to the nature of the damage is carried out by visual analysis. Thus, in the case of destruction of the cell wall, low flocculent areas are formed on the surface. In the case of destruction of the polysaccharide capsule, long and high fibrous structures are formed on the surface. A clarification on this issue has been added to the text of the article.
Point 5: Fig. 3B is presented in the form of a table, and the expression of the results is not intuitive enough. It is suggested to present the results with figures and supplemented with explanations.
Response 5: Now the data is presented in the form of a figure and supplemented with an explanation.

Round 2
Reviewer 3 Report
-
Reviewer 4 Report
The manuscript has been sufficiently improved to warrant publication in Cells.